# Evaluating indirect genetic effects of siblings using singletons

**Laurence J. Howe**[1,2]*, **David M. Evans**[1,3,4], **Gibran Hemani**[1,2], **George Davey Smith**[1,2], **Neil M. Davies**[1,2,5]*

**1** Medical Research Council Integrative Epidemiology Unit, Population Health Sciences, University of Bristol, Bristol, United Kingdom, **2** Population Health Sciences, Bristol Medical School, University of Bristol, Bristol, United Kingdom, **3** University of Queensland Diamantina Institute, University of Queensland, Brisbane, Australia, **4** Institute for Molecular Bioscience, University of Queensland, Brisbane, Australia, **5** K.G. Jebsen Center for Genetic Epidemiology, Department of Public Health and Nursing, NTNU, Norwegian University of Science and Technology, Trondheim, Norway

* laurence.howe@bristol.ac.uk (LJH); Neil.Davies@bristol.ac.uk (NMD)

## Abstract

Estimating effects of parental and sibling genotypes (indirect genetic effects) can provide insight into how the family environment influences phenotypic variation. There is growing molecular genetic evidence for effects of parental phenotypes on their offspring (e.g. parental educational attainment), but the extent to which siblings affect each other is currently unclear. Here we used data from samples of unrelated individuals, without (singletons) and with biological full-siblings (non-singletons), to investigate and estimate sibling effects. Indirect genetic effects of siblings increase (or decrease) the covariance between genetic variation and a phenotype. It follows that differences in genetic association estimates between singletons and non-singletons could indicate indirect genetic effects of siblings if there is no heterogeneity in other sources of genetic association between singletons and non-singletons. We used UK Biobank data to estimate polygenic score (PGS) associations for height, BMI and educational attainment in self-reported singletons (N = 50,143) and non-singletons (N = 328,549). The educational attainment PGS association estimate was 12% larger (95% C.I. 3%, 21%) in the non-singleton sample than in the singleton sample, but the height and BMI PGS associations were consistent. Birth order data suggested that the difference in educational attainment PGS associations was driven by individuals with older siblings rather than firstborns. The relationship between number of siblings and educational attainment PGS associations was non-linear; PGS associations were 24% smaller in individuals with 6 or more siblings compared to the rest of the sample (95% C.I. 11%, 38%). We estimate that a 1 SD increase in sibling educational attainment PGS corresponds to a 0.025 year increase in the index individual's years in schooling (95% C.I. 0.013, 0.036). Our results suggest that older siblings may influence the educational attainment of younger siblings, adding to the growing evidence that effects of the environment on phenotypic variation partially reflect social effects of germline genetic variation in relatives.

information or go to https://www.ukbiobank.ac.uk/enable-your-research/apply-for-access. Summary data from the within-sibship GWAS is available from https://gwas.mrcieu.ac.uk/ Code for simulations is available at SiblingIGE/simulations.R at main · LaurenceHowe/SiblingIGE (github.com).

**Funding:** LJH, GH, GDS and NMD work in a unit that receives support from the University of Bristol and the UK Medical Research Council (MC_UU_00011/1). DME is supported by an NHMRC Senior Research Fellowship (APP1137714). The funders had no role in study design, data collection and analysis, decision to publish, or preparation of the manuscript.

**Competing interests:** The authors have declared that no competing interests exist.

## Author summary

Genetic data from families can be used to evaluate social effects of parents on their offspring. For example, non-transmitted parental genetic variants have been shown to associate with offspring educational attainment indicative of parental effects. Siblings may also influence one another but available data has limited our understanding of sibling effects.

Associations between genetic variants and phenotypes in the same individual will capture effects of sibling genetic variants because of the genomic similarity between siblings. We propose that sibling effects can be evaluated by comparing genetic association estimates between singletons (no siblings), who cannot be plausibly influenced by sibling effects, and non-singletons (one or more siblings).

We apply this approach to data from a large population biobank with follow-up analyses investigating effects of birth order. We find evidence of sibling effects on educational attainment, but not on height or body mass index, with our results suggesting that older siblings influence the educational attainment of younger siblings.

## Introduction

Parents transmit genetic variation to their offspring and shape their early-life environment [1,2]. Parental effects on the family environment partially relate to effects of parental genotypes (indirect genetic effects). For example, parental genotypes which influence their behaviour could impact the offspring's environment [1,3,4] Other relative classes such as siblings may also have indirect effects on their relatives [5]; older siblings could influence the school achievement of younger siblings [6] or their smoking behaviour [7].

Phenotypic data from twins (monozygotic/dizygotic), foster siblings and only children can be used to estimate sibling effects by considering differences in phenotypic variance [8,9]. A complementary approach is to use molecular genetic data from extended families to estimate indirect genetic effects of parents and siblings [1,4,10,11,12–14], which can then inform the effects of parental and sibling phenotypes. However, existing studies that have sampled family members have limited power to estimate sibling effects because of the paucity of available family data and the statistical inefficiency of within-family models, even when genotypes of missing first degree relatives are imputed [12,13,15]. Here, we propose an alternative molecular genetics approach for evaluating indirect genetic effects of siblings which can use large samples of unrelated individuals and so is likely to have higher statistical power than within-family approaches.

In a population sample of unrelated individuals, the association between a genetic variant and a phenotype reflects the effect of the genetic variant (or a correlated variant) on the phenotype in the index individual (direct effect), indirect genetic effects of relatives which strengthen (or weaken) the genotype-phenotype covariance, as well as demography (assortative mating, population stratification) [4,16,17]. One approach to evaluate indirect genetic effects is to compare genetic associations between non-adopted and adopted individuals. Adopted individuals are raised apart from their biological families so genetic associations will not capture indirect genetic effects of relatives [11,18]. Extending this intuition, we note that indirect genetic effects of siblings will not impact genetic association estimates from singletons (individuals without siblings). If other sources of genetic association are consistent between singletons and non-singletons, then differences between the singleton and non-singleton genetic association estimates could indicate indirect genetic effects of siblings. Potential group-level differences

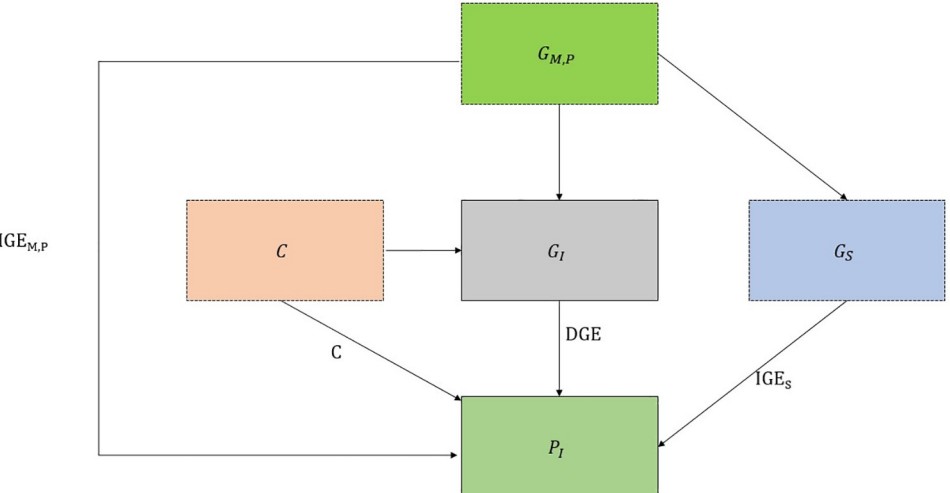

**Fig 1. Sources of association between genotypes and phenotypes.** The association between an individual's genotype ($G_I$) and a phenotype ($P_I$) will capture the following: 1) direct genetic effects (DGE) of inheriting a variant or a correlated variant. 2) indirect genetic effects ($IGE_{M,P,S}$) of parental (maternal/paternal $G_{M,P}$) and sibling genotypes ($G_S$) on $P_I$ via the shared environment because of the correlations between $G_I$, $G_{M,P}$ and $G_S$. 3) Confounding factors (C) such as population stratification and assortative mating which confound the associations between genetic variants and phenotypes. If an individual has no siblings (or is raised apart from their siblings) then the $G_I$, $P_I$ association cannot, by definition, be affected by indirect genetic effects of siblings. Therefore, indirect genetic effects of siblings are a possible explanation for heterogeneity in genetic associations between singletons and non-singletons. Note that this figure features several simplifications such as maternal and paternal indirect genetic effects being consistent and no paths between confounding factors and parental or sibling genotypes.

between singletons and non-singletons are unlikely to confound genetic associations unless the factor can influence genotype (e.g. ancestry) (**Fig 1**).

## Results

### PGS associations in singletons and non-singletons

To evaluate potential indirect genetic effects of siblings, we explored differences in genetic association estimates between singletons and non-singletons for height, BMI and educational attainment genetic variants. We constructed polygenic scores (PGS) for these phenotypes using Genome-wide Association Study (GWAS) summary data independent of UK Biobank [19]. We estimated associations between the PGS and the relevant phenotype in the singleton and non-singleton samples, adjusting for sex, birth year and the first 10 ancestry informative principal components. We then estimated the difference (% attenuation) from the non-singleton to the singleton PGS association estimate for each phenotype.

The non-singleton educational attainment PGS association estimate was 12% larger than the singleton estimate (95% C.I. 3%, 21%; difference P = 0.009). In contrast, we found no strong evidence for differences between the singleton and non-singleton PGS estimates for height (attenuation = 0%; 95% C.I. -3%, 2%) and BMI (attenuation = 5%; 95% C.I. -1%, 11%) (**Fig 2** and **Table A in S1 Text**).

As a sensitivity analysis, we repeated the educational attainment analysis using PGS weightings from a recent within-sibship GWAS meta-analysis of educational attainment [20]. The within-sibship weighted PGS is unlikely to capture effects of population stratification, assortative mating and indirect genetic effects [3,4,20]. However, associations between this PGS and educational attainment in our sample (of unrelated individuals) may still be affected by these

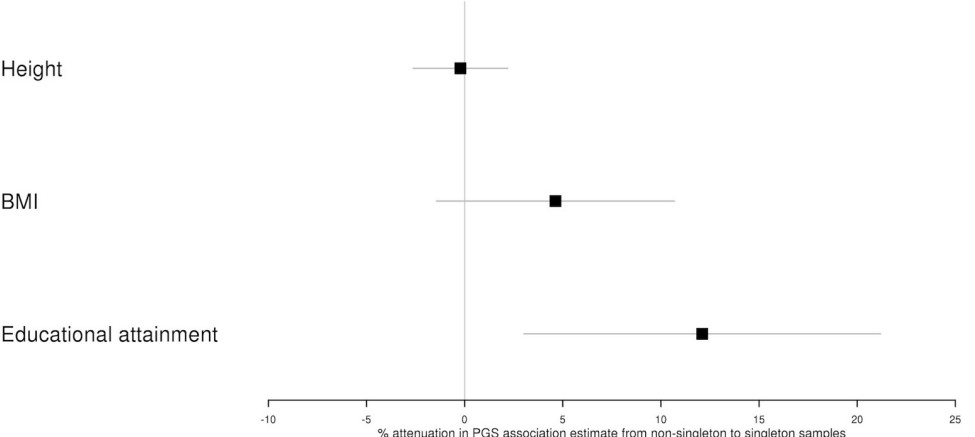

**Fig 2. Differences between singleton and non-singleton PGS estimates.** Fig 2 illustrates the % attenuation in PGS association estimates from non-singletons to singletons for height, BMI and educational attainment.

sources of association. This is because genetic variants in the within-sibship PGS with direct genetic effects on educational attainment may also have non-direct genetic effects which are not controlled for in models that don't account for parental genotypes. The singleton within-sibship attenuation in the educational attainment PGS association estimate (11%; 95% C.I. -1%, 23%) was highly consistent with the estimate from the primary analysis (12%; 95% C.I. 3%, 21%) (**Table A in S1 Text**).

## PGS associations and birth order

Birth order may influence the magnitude of sibling effects. For example, the schooling decisions of an older sibling are more likely to influence a younger sibling than the converse. We used birth order data (available in a subset of the UK Biobank cohort) to identify firstborns (individuals with only younger siblings) and non-firstborns (individuals with one or more older siblings). We then computed PGS associations, as above, in the firstborn and non-firstborn samples.

PGS educational attainment association estimates were larger in non-firstborns than in firstborns (attenuation = 14%, 95% C.I. 3%, 26%, difference P = 0.013) with no strong evidence of heterogeneity for height or BMI. The firstborn PGS educational attainment association estimate was highly consistent with the singleton estimate (difference P = 0.93), suggesting that the larger PGS association in non-singletons is due to individuals with older siblings, rather than firstborns (**Fig 3 and Table B in S1 Text**).

## PGS associations for education by number of siblings

We next evaluated evidence for a linear relationship between number of siblings and the association of the educational attainment PGS with measured educational attainment. In the full sample (N = 378,445), we found no strong evidence for an interaction (P = 0.24). However, this sample included very large families where data could be more susceptible to misclassification error and the association estimates may be affected by lower parental investment or other confounding factors. Indeed, we found that the PGS association estimate from 18,746 individuals self-reporting having 6 or more siblings was 24% smaller than the estimate from individuals with 5 or fewer siblings (95% C.I. 11%, 38%, P = 5.2x10$^{-4}$) (**Fig 4**).

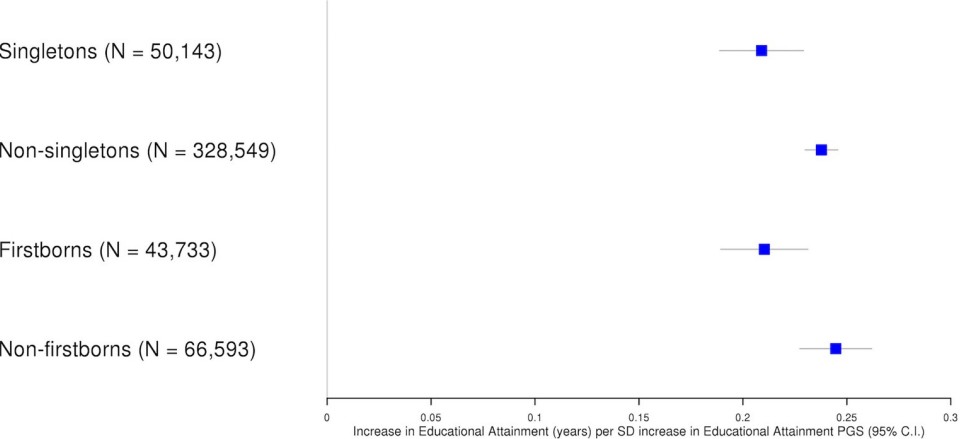

**Fig 3. Educational attainment PGS and years in schooling by singleton status and birth order.** Fig 3 displays association estimates between educational attainment PGS and educational attainment in singletons, non-singletons, firstborns and non-firstborns.

We evaluated whether there is a non-linear relationship between number of siblings and the PGS associations by applying a quadratic model including the square of the number of siblings and a quadratic interaction term. This model provided evidence of a non-linear relationship with the linear interaction estimate indicating a 0.017 increase in PGS association estimate per each additional sibling (95% C.I. 0.008, 0.025; P = 0.0001) in the opposite direction to the quadratic interaction estimate which indicated a 0.002 decrease in PGS association estimate per each unit increase in the square of the number of siblings (95% C.U. -0.001, -0.003; P = 0.0003).

We also repeated the linear analysis after removing outlying individuals with extreme values, considering individuals with 6 or more siblings as outliers based on an outlier threshold of 5% because this group corresponded to 4.9% of the total sample. In this sample of individuals with 5 or fewer siblings we found evidence for a linear relationship; each additional sibling corresponded to an increase of 0.012 in the PGS association estimate (95% C.I. 0.006, 0.018;

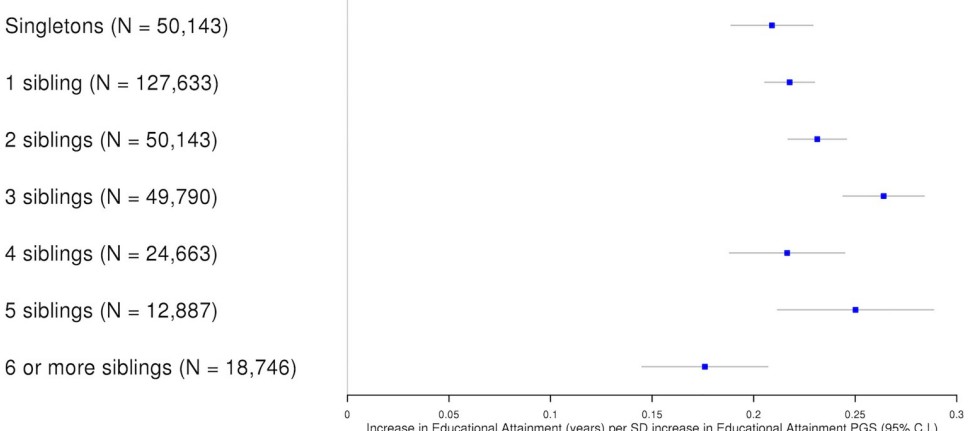

**Fig 4. Educational attainment PGS and years in schooling by number of siblings.** Fig 4 displays association estimates between educational attainment PGS and educational attainment stratified by the number of self-reported siblings. Individuals reporting 6 or more siblings were combined into the same category.

P = 4.3x10$^{-5}$). Under assumptions discussed in the **Methods**, this estimate can be scaled (multipled by two) to provide an estimate of sibling indirect genetic effects; a 1 SD increase in the educational attainment PGS of a sibling increases an index individual's years in schooling by 0.025 years (0.013, 0.036). For comparison, this estimate is 11% (95% C.I. 5%, 16%) of the magnitude of the PGS association estimate in the full sample (0.23 years, 95% C.I. 0.23, 0.24).

### Phenotypic differences between singletons and non-singletons

As discussed in the **Methods**, comparisons of genetic association estimates between singletons and non-singletons may be sensitive to group-level differences. We compared the sex, age, height, BMI, measured educational attainment and educational attainment PGS of singletons and non-singletons. Singletons were more likely to be male (+1.0%; 95% C.I. 0.6%, 1.5%), older (+2.6 years; 95% C.I. 2.6, 2.7) and were born further south (5.2 km; 95% C.I. 3.6, 6.8) and east (3.5km; 95% C.I. 2.7, 4.3). After adjusting for age and sex, we found evidence that singletons are taller (+0.15 cm; 95% C.I. 0.09, 0.21), have higher BMI (+0.06 kg/m$^2$; 95% C.I. 0.01, 0.10) and have more years in full-time education (+0.25 years; 95% C.I. 0.23, 0.27). However, we found no strong evidence of differences between singletons and non-singletons for educational attainment PGS (-0.002 SD difference; 95% C.I. -0.011, 0.008) (**Tables C and D in S1 Text**). Similar differences were also observed between singletons and firstborn non-singletons for age and BMI but contrastingly, singletons were shorter (-0.23 cm; -0.31, -0.14) and had fewer years in full-time education (-0.09 years; 95% C.I. -0.12, -0.06) than firstborns (**Table E in S1 Text**). These findings illustrate differences between singletons and non-singletons, which could relate to parental differences (e.g. education) but also birth order effects or factors influencing study participation. However, we note that group-level differences by family size are unlikely to confound our genetic association analyses with the exception of ancestral differences and that some of the observed differences are relatively modest (e.g. 5.2 km difference in birth coordinates).

## Discussion

Here, we proposed that differences in genetic association estimates between singletons and non-singletons can be used to evaluate indirect genetic effects of siblings. Using UK Biobank data, we found that the association between the educational attainment PGS and educational attainment was larger for non-singletons than singletons. This difference was driven by individuals with older siblings rather than firstborns. We found that the relationship between number of siblings and educational attainment PGS associations was non-linear, with PGS associations attenuating substantially in larger families with more than 6 children. After removing these families, we found strong evidence for a linear relationship between number of siblings and the educational attainment PGS associations. These findings are suggestive of indirect genetic effects of siblings; older siblings influence the education of younger siblings.

There are alternative explanations for our findings. First, group-level differences between singletons and non-singletons, such as for parental education [21] or health outcomes [22–24], could have led to differences in sources of genetic association [25]. For example, direct and indirect genetic effects on educational attainment could vary by socio-economic position [26–28] or by other covariates. Indeed, we observed differences between singletons and non-singletons for age, sex, and several phenotypes, consistent with parental differences, birth order effects or selection bias although we did not observe differences for the educational attainment PGS. Second, our results could have been impacted by collider bias [25,29], because stratifying on number of siblings (which is non-random) could distort associations between factors which influence number of siblings. Third, indirect genetic effects of parents may be stronger

in singletons because of additional parental investment with fewer offspring. Similarly birth order may influence the magnitude of direct genetic effects and how an individual is affected by their parents [30,31]. However, differences in indirect genetic effects and birth order are unlikely to explain the observed differences in PGS associations for educational attainment. Previous literature [32] suggests that singletons and firstborns are more likely to receive additional parental investment, which would likely result in larger indirect genetic effects of parents on educational attainment and a larger PGS association estimate. Inconsistent with this explanation, we observed larger PGS estimates in non-singletons and non-firstborns. A more plausible explanation for the difference in association of education and the educational attainment PGS between firstborns and non-firstborns is that older siblings influence the educational decisions of younger siblings. Whereas an individual's decision to go to university is less likely to be strongly influenced by younger siblings.

Previous research has shown that PGS-phenotype associations can differ across ancestry groups as well as by other phenotypes such as socio-economic position, age and sex [27,33]. Indeed, it has been previously demonstrated that educational attainment PGS more strongly predict educational attainment in individuals with one sibling compared to individuals with no siblings, although these findings were not interpreted with respect to sibling IGEs or birth order [27]. As a sensitivity analysis, we performed analyses using an educational attainment PGS weighted by within-sibship GWAS estimates [20] which are robust against population stratification, assortative mating and indirect genetic effects of parents but not indirect genetic effects of siblings. Here we found consistent evidence that PGS association estimates are larger in non-singletons. This suggests that our results are unlikely to be explained by effects of population stratification, assortative mating and indirect genetic effects on the PGS weightings. However, these mechanisms could still affect the association between the PGS and educational attainment differently between singletons and non-singletons because analyses in unrelated individuals do not account for variance in parental genotypes.

Molecular genetic analyses of indirect genetic effects of relatives [1,4,11,12,14] have generally found evidence of imitation rather than contrast effects [8,9,34], i.e. effects of parental genotypes on the shared environment result in children being more similar to their parents. Our results are consistent with imitation effects of siblings for educational attainment as they suggest stronger rather than weaker gene-environment correlations for individuals with older siblings. For example, this could occur if an older sibling going to university increases the probability that their younger sibling will also go to university. Contrastingly, there could be more subtle effects of an individual's behaviour on the shared family environment. For example, an individual with a higher education PGS may help younger siblings more with their homework.

A previous meta-analysis estimated that parental indirect genetic effects on educational attainment are around half of the magnitude of direct genetic effects of inherited variants [14]. We were unable to estimate direct genetic effects in this study but estimated that the indirect genetic effects of one sibling on education are around a tenth of the total genetic association estimate suggesting that sibling indirect genetic effects on education are likely to be substantially smaller than parental effects. However, there are complexities specific to interpreting estimates of indirect genetic effects of siblings. First, our estimate was derived using all non-singletons, but birth order is likely to affect the magnitude of indirect genetic effects of siblings. For example, supported by our findings, sibling indirect genetic effects could be much larger for non-firstborns. Second, we assumed a linear additive model with each additional sibling increasing the combined magnitude of the sibling indirect genetic effects. However, we observed an attenuated genetic association estimate in larger families suggesting that the relationship between number of siblings and genetic associations may be non-linear. The genetic

association attenuation in larger families could also relate to differences in indirect genetic effects of parents or confounding.

In contrast to our findings, previous studies using family-based approaches have reported limited evidence of sibling IGEs [12,13,18]. This could have been because of differences in statistical power or because of genuine heterogeneity. We compared our sibling IGE estimates to the estimates from one of these manuscripts (Kong et al [13]) and found limited evidence of heterogeneity. Our sibling IGE estimate, which was more precisely estimated, was consistent with the estimate and 95% confidence interval from Kong et al [13] (**S1 Text**). This suggests that the differences in conclusions are likely to relate to differences in statistical power.

Our findings add to the growing evidence for social effects of germline genetic variation. The main limitation of our approach is that it is sensitive to systematic differences between singletons and non-singletons. For example, interactions between the PGS and covariates could have biased our estimates of sibling indirect genetic effects. An additional limitation is that (beyond including birth year as a covariate) we did not account for possible generational effects (i.e. effects of the PGS changing over time) which could induce bias in combination with changes in family size over time. Further caveats are that our analyses may have been affected by selection bias relating to non-random participation in UK Biobank [29,35], that we did not account for possible indirect genetic effects of half siblings in our analyses as this data was not available and that we assumed random mating with assortative mating a likely source of bias. We also did not evaluate whether sibling indirect genetic effects vary by the sex of the index individual and their siblings (e.g. same sex sibling pairs may be more likely to influence one-another). Larger datasets of first-degree relatives will enable more precise estimation of sibling indirect genetic effects and allow the evaluation of sibling effects on a wider range of phenotypes such as smoking behaviour and alcohol consumption.

## Methods

### Ethics statement

UK Biobank received ethical approval from the Research Ethics Committee (11/NW/0383). Access to UK Biobank data was granted as part of application 8786 (PI: NMD).

### Evaluating indirect genetic effects of siblings using singletons

There are several different mechanisms which can induce covariance between an individual's genotype $G_I$ and their trait value $Y$ ($Cov[G_I, Y]$).

a. Direct genetic effects, which we define as the effect of inheriting $G_I$ on $Y$. For simplicity, we assume that the genetic variant is causal even though variants identified in GWAS are usually in linkage disequilibrium with the true causal variants.

b. Indirect genetic effects of the maternal ($G_M$), paternal ($G_P$) and sibling ($G_S$) genotype on $Y$ via the shared environment. These effects are partially captured by $Cov[G_I, Y]$ because an individual's genotype is correlated with the genotype of their relatives (i.e. indirect genetic effects of relatives induce gene-environment correlation). Note that effects of other relatives (e.g. grandparents) could also be captured by $Cov[G_I, Y]$ and the formulation below could be extended to model these effects.

c. Effects of unmeasured confounders ($C$) which influence both $G_I$ and $Y$ (e.g. population stratification, assortative mating).

We consider how indirect genetic effects of siblings would affect the covariance in non-singletons ($Cov_{NS}[G_I, Y]$) who have siblings. We model $Y$ as a function of genotypes ($G_{I,M,P,S}$) and

error variable $\in$ (with mean 0). We assume a single autosomal genetic variant, additive genetic effects only, random mating, no confounding between $G_I$ and $Y$ and that non-singletons have one sibling each.

$$Y = k_I G_I + k_M G_M + k_P G_P + k_S G_S + \in$$

where $k_{I,M,P,S}$ are the (direct/indirect) effects of $G_{I,M,P,S}$ on $Y$

It follows that the population covariance of individuals with siblings $Cov_{NS}[G_I, Y]$ is:

$$Cov_{NS}[G_I, Y] = (k_{NS,I} + 0.5k_{NS,M} + 0.5k_{NS,P} + 0.5k_{NS,S})Var[G_I]$$

Further detail on the derivation of this covariance term is contained in **S1 Text**.

In contrast, in singletons, where indirect genetic effects from siblings are not possible, $k_S$ would equal zero and the population covariance ($Cov_S[G_I, Y]$) would simplify to:

$$Cov_S[G_I, Y] = (k_{S,I} + 0.5k_{S,M} + 0.5k_{S,P})Var[G_I]$$

The ordinary least squares (OLS) regression estimates of $G_I$ on $Y$ will capture indirect genetic effects of siblings. It follows that the expected OLS non-singleton $\beta_{NS}$ and singleton $\beta_S$ regression coefficients are:

$$\hat{\beta}_{NS} = (k_{NS,I} + 0.5k_{NS,M} + 0.5k_{NS,P} + 0.5k_{NS,S})$$

$$\hat{\beta}_S = (k_{S,I} + 0.5k_{S,M} + 0.5k_{S,P})$$

As above, we assumed that each index individual had only one sibling. However, the model could be extended to account for different numbers of siblings ($N$). For example, assuming a linear relationship:

$$\hat{\beta}_{NS} = (k_{NS,I} + 0.5k_{NS,M} + 0.5k_{NS,P} + (N * 0.5k_{NS,S}))$$

Dependent on certain assumptions, twice the difference between $\beta_{NS}$ and $\beta_S$ can be used to provide an unbiased estimate of indirect genetic effects of siblings.

$$2(\hat{\beta}_{NS} - \hat{\beta}_S) = k_S$$

Assumption 1: No heterogeneity in direct effects ($k_I$) between non-singletons and singletons.

$$k_{NS,I} = k_{S,I}$$

If this assumption does not hold then the difference estimator will be biased by the difference in direct effects ($k_{NS,I} - k_{S,I}$).

$$\hat{\beta}_{NS} - \hat{\beta}_S = k_S + (k_{NS,I} - k_{S,I})$$

Assumption 2: No heterogeneity in maternal ($k_M$) or paternal ($k_P$) indirect effects between non-singletons and singletons.

$$k_{NS,M} = k_{S,M}$$

$$k_{NS,P} = k_{S,P}$$

If this assumption does not hold then the difference estimator will be biased by half of the

differences in maternal and paternal indirect effects ($k_{NS,M}–k_{S,M}$, $k_{NS,P}–k_{S,P}$).

$$\hat{\beta}_{NS} - \hat{\beta}_S = k_S + 0.5(k_{NS,M} - k_{S,M}) + 0.5(k_{NS,P} - k_{S,P})$$

Assumption 3: No heterogeneity in confounders of $G_I$ and $Y$ between non-singletons and singletons.

If this assumption does not hold then the difference estimator will be biased by the confounder $C$ which must influence both $G_I$ and $Y$ by definition so must be a function of $G_I$ in the context of $Y$.

$$\hat{\beta}_{NS} - \hat{\beta}_S = k_S + C'$$

where $C'$ is a derivative of $C$.

If these assumptions hold, then twice the difference between the two regression coefficients will be an unbiased estimator of the indirect genetic effects of siblings.

We note that group-level differences between singletons and non-singletons could lead to violations of the three assumptions. Indeed, there are well-observed group-level differences between singletons and non-singletons. For example, higher education is associated with having fewer children, so parents of singletons are likely to be more educated. However, group-level differences will not necessarily lead to differences in genetic association estimates.

Genetic differences (e.g. mean or variance) between singletons and non-singletons combined with non-linear effects or interactions could lead to violation of all three assumptions. The second assumption would be violated if indirect genetic effects of parents are stronger (or weaker) for singletons because of differences in parental investment. The third assumption could be violated if there are differences in assortative mating and population structure between singletons and non-singletons.

A further caveat is that stratifying on a non-random variable (e.g. number of siblings) could distort associations between determinants of $Y$ (collider bias) [29]. In this context, collider bias is unlikely to have a large effect on genetic associations. Although number of siblings is non-random, it is unlikely to be directly influenced by $G_I$ or $Y$. However, collider bias could distort estimates of $\beta_{NS}$ and $\beta_S$ via paths involving parental characteristics that influence the number of siblings of the index individual. We note that if the effects of collider bias on $\beta_{NS}$ and $\beta_S$ are consistent, then this bias would cancel out in the difference.

## Simulations

To explore properties of the proposed framework we performed simulations. First, we confirmed the theory from above that sibling indirect genetic effects will lead to differences in genetic associations between singletons and non-singletons in simulated data. Second, we evaluated how non-random mating affects parameter estimates from the framework. Code for simulations is available at SiblingIGE/simulations.R at main · LaurenceHowe/SiblingIGE (github.com).

**Model 1 –Random mating.** In Model 1, we simulated 100,000 parent-offspring trios with singleton offspring and 100,000 parent-offspring quads with two non-singleton offspring (i.e. a sibling pair).

a) Parent-offspring trios

We simulated a normally distributed PGS for each individual using the following variance-covariance matrix to characterise within-family correlations for maternal ($PGS_M$), paternal

($PGS_P$), and offspring ($PGS_O$) PGS under random mating.

$$M \quad P \quad O$$

$$
\begin{matrix}
1 & 0 & 0.5 \\
0 & 1 & 0.5 \\
0.5 & 0.5 & 1
\end{matrix}
$$

The offspring phenotype $Y_O$ was simulated as a function of the direct effect of $PGS_O$, indirect effects of $PGS_M$ and $PGS_P$, effects of confounding $C$ and a normally distributed error term ($\in \sim N(0, 1)$).

$$Y_O = k_O PGS_O + k_M PGS_M + k_P PGS_P + (C*PGS_O) + \in$$

where $k_{O,M,P}$ are the (direct/indirect) effects of $PGS_{O,M,P}$ on $Y_O$.

b) Parent-offspring quads

We simulated a normally distributed PGS as above for the parent-offspring trios but with two offspring using the following variance-covariance matrix.

$$M \quad P \quad O_1 \quad O_2$$

$$
\begin{matrix}
1 & 0 & 0.5 & 0.5 \\
0 & 1 & 0.5 & 0.5 \\
\\
0.5 & 0.5 & 1 & 0.5 \\
0.5 & 0.5 & 0.5 & 1
\end{matrix}
$$

The phenotype of $O_1$ ($Y_{O_1}$) was simulated as above but including an additional effect of the PGS of their sibling $O_2$.

$$Y_{O_1} = k_{O_1} PGS_{O_1} + k_M PGS_M + k_P PGS_P + k_{O_2} PGS_{O_2} + C(PGS_{O_1}) + \in$$

where $k_{O_1,M,P,O_2}$ are the (direct/indirect) effects of $PGS_{O_1,M,P,O_2}$ on $Y_{O_1}$.

We then computed the OLS regression estimates of an individual's own PGS on their phenotype, i.e., $PGS_O$ on $Y_O$ in the singleton sample and $PGS_{O_1}$ on $Y_{O_1}$ in the non-singleton sample.

Simulations confirmed the theoretical expectations from above that the OLS estimates in the singleton $\hat{\beta}_S$ and non-singleton $\hat{\beta}_{NS}$ samples are as follows:

$$\hat{\beta}_S = (k_O + 0.5k_M + 0.5k_P + C)$$

$$\hat{\beta}_{NS} = (k_{O_1} + 0.5k_M + 0.5k_P + 0.5k_{O_2} + C)$$

The simulations also confirmed the expectations of bias relating to violation of the assumptions and that the difference estimator $\hat{\beta}_{NS} - \hat{\beta}_S$ provides an unbiased estimate of sibling indirect genetic effects if all the assumptions hold.

**Model 2 –Non-random mating.** In Model 2, we extended Model 1 by modifying the variance-covariance matrix of the within-family PGS to reflect assortative mating. This involved increasing the maternal-paternal correlations from 0 to 0.2 and the parent-offspring and sibling correlations from 0.5 to 0.6. Phenotypes were simulated as in Model 1.

a) Parent-offspring trios

$$M\ P\ O$$

$$
\begin{matrix}
1 & 0.2 & 0.6 \\
0.2 & 1 & 0.6 \\
0.6 & 0.6 & 1
\end{matrix}
$$

b) Parent-offspring trios

$$M\ P\ O_1\ O_2$$

$$
\begin{matrix}
1 & 0.2 & 0.6 & 0.6 \\
0.2 & 1 & 0.6 & 0.6 \\
0.6 & 0.6 & 1 & 0.6 \\
0.6 & 0.6 & 0.6 & 1
\end{matrix}
$$

In Model 2, the OLS regression estimates $\hat{\beta}_S$ and $\hat{\beta}_{NS}$ were slightly higher than in Model 1 because of the increased covariance of the PGS within-families. Notably, the difference term $(\hat{\beta}_{NS} - \hat{\beta}_S)$ was also inflated over expectation suggesting that assortative mating, if not accounted for, could lead to overestimates of sibling indirect genetic effects from the proposed framework.

## UK Biobank

UK Biobank is a large prospective cohort study of 503,325 individuals, aged between 38–73 years at baseline, who were recruited between 2006 and 2010 from across the United Kingdom. UK Biobank study participants were genotyped, completed questionnaire data at baseline and have linked records with secondary care data and other health registries. The cohort has been described in detail in previous publications, including information on genotyping [19,36].

In this study, we used UK Biobank genetic data and phenotype data (height, BMI, educational attainment, number of siblings, adoption status and birth order). Height of study participants was measured at baseline using a Seca 202 device at the assessment centre (field ID: 12144–0.0). BMI was derived manually using measures of standing height and weight (field ID: 21001.0.0). Educational attainment was defined using qualification data, as in a previous study [37]. Questionnaire data included information on the highest level of educational attainment which was then used to estimate the number of years spent in full-time education (field ID: 6138). Individuals were asked the number of full brothers (field ID: 1873), full sisters (field ID: 1883) and older siblings (field ID: 5057) in the baseline questionnaire. Individuals were also asked if they were adopted as a child (field ID: 1767). North-south (field ID: 129) and East-West (field ID: 130) birth coordinates were derived from questionnaire data on place of birth.

Starting with the full UK Biobank dataset with genetic data, we restricted analyses to individuals of recent European descent based on a k-means cluster analysis on the first 4 genetic principal components and also removed closely-related individuals and standard exclusions (e.g. sex mismatch). More information on the internal quality control of UK Biobank data is contained in a previous document [38]. This dataset included 385,222 individuals. For the

purposes of our analyses, we then removed 6,025 individuals who reported being adopted as a child.

We defined singletons (N = 50,143) as individuals who self-reported having both no full brothers and no full sisters. We defined non-singletons (N = 328,549) as individuals who self-reported having 1 or more full brothers or 1 or more full sisters. Individuals who self-reported zero brothers or sisters to one question but did not answer the other question were set to missing. Data on birth order was available in a subset of the cohort (N = 110,326). Firstborns (N = 43,733) were defined as individuals who self-reported no older siblings while non-firstborns (N = 66,593) reported one or more older siblings.

## PGS construction and association analyses

We constructed PGS for height, BMI and educational attainment by LD clumping ($P < 1.0 \times 10^{-5}$, $r^2 < 0.001$, clumping distance = 10000 kb) summary data from previous GWAS. We used data from GWAS independent of UK Biobank for height [39], BMI [40] and educational attainment [37] to minimise sample overlap. For educational attainment, we used the summary data from the discovery sample which included 23AndMe. As a sensitivity analysis, we also constructed educational attainment PGS using weightings from a recent within-sibship GWAS of educational attainment [20], i.e. we included the same variants in the PGS but used the beta values from the within-sibship GWAS. This approach would reduce effects of population stratification, assortative mating and indirect genetic effects on the PGS itself. However, these sources of genetic association could still impact the association between the within-sibship PGS and educational attainment.

We estimated associations between PGS and the relevant phenotype (e.g. height PGS and height) in the singleton, non-singleton, firstborn and non-firstborn samples separately, adjusting for sex, birth year and the first 10 ancestry-informative principal components. We estimated the % attenuation between the non-singleton and singleton PGS estimates using the difference of two means approach (based on the delta method) [41] and then rescaled the difference term and confidence interval as a ratio term (i.e. % attenuation = difference/non-singleton estimate). The estimates were derived in non-overlapping samples, so we assumed zero covariance.

To evaluate a linear relationship between number of siblings and the educational attainment PGS estimate, we applied the following regression model including an interaction term (number of siblings multiplied by the PGS) to estimate the effect of each additional sibling on the PGS association estimate. In the first instance, we used the full sample of individuals with complete data on education and number of siblings.

$$Education \sim PGS + N + (N * PRS) + BirthYear + Sex + PC1\ldots + PC10$$

where $PGS$ = educational attainment PGS, $N$ = number of siblings.

To evaluate a non-linear relationship, we estimated associations between educational attainment PGS and educational attainment separately in individuals with differing numbers of siblings (0, 1, 2, 3, 4, 5, 6 or more). We compared PGS estimates between individuals with 6 or more siblings and the rest of the sample using the difference of two means approach.

We extended the linear model with a quadratic term including the number of siblings squared.

$$Education \sim PGS + N + (N * PRS) + N^2 + (N^2 * PRS)BirthYear + Sex + PC1\ldots + PC10$$

where $PGS$ = educational attainment PGS, $N$ = number of siblings.

We also repeated the linear relationship regression model after removing 18,746 outlier individuals with 6 or more siblings (4.9% of the sample). We multiplied the regression coefficient for the interaction term by two to generate an estimate of the indirect genetic effect of one sibling.

## Phenotypic differences between singletons and non-singletons

We explored group-level difference between singletons and non-singletons for sex, birth year, birth coordinates (north-south, east-west), height, BMI, educational attainment and an educational attainment PGS. First, we extracted information on the mean and standard deviations for each of these measures for singletons, non-singletons, firstborns and non-firstborns. Second, we tested for differences in sex, birth year and birth coordinates (north-south, east west) using the difference of two means approach (based on the delta method) [41]. Third, we used a linear regression adjusting for sex and age with either (singleton 0, non-singleton 1) or (firstborn 0, non-firstborn 1) to investigate group-level differences for height, BMI, educational attainment and the educational attainment PGS.

## Supporting information

**S1 Text. Supplementary Materials. Table A in S1 Text.** PGS association estimates for singletons and non-singletons. Table displays association estimates between height, BMI and educational attainment PGS with the same phenotype in singletons and non-singletons. **Table B in S1 Text.** PGS association estimates for firstborns and non-firstborns. Table displays association estimates between height, BMI and educational attainment PGS with the same phenotype in firstborns and non-firstborns. **Table C in S1 Text.** Characteristics of singletons, non-singletons, firstborns and non-firstborns in UK Biobank. Table contains descriptives of group-level characteristics of singletons, non-singletons, firstborns and non-firstborns. **Table D in S1 Text.** Differences between singletons and non-singletons. Table contains estimates of differences in group-level characteristics between singletons and non-singletons. **Table E in S1 Text.** Differences between singletons and firstborns. Table contains estimates of differences in group-level characteristics between firstborns and non-firstborns.
(DOCX)

## Author Contributions

**Conceptualization:** Laurence J. Howe.

**Formal analysis:** Laurence J. Howe.

**Funding acquisition:** George Davey Smith.

**Investigation:** Laurence J. Howe, David M. Evans, Gibran Hemani, George Davey Smith, Neil M. Davies.

**Methodology:** Laurence J. Howe.

**Supervision:** David M. Evans, Gibran Hemani, George Davey Smith, Neil M. Davies.

**Writing – original draft:** Laurence J. Howe.

**Writing – review & editing:** Laurence J. Howe, David M. Evans, Gibran Hemani, George Davey Smith, Neil M. Davies.

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
