## [Decision Letter · Decision Letter 0]

2 Sep 2021

Dear Dr Howe,

Thank you very much for submitting your Research Article entitled 'Evaluating indirect genetic effects of siblings using singletons' to PLOS Genetics.

The manuscript was fully evaluated at the editorial level and by independent peer reviewers. The reviewers appreciated the attention to an important problem, but raised some substantial concerns about the current manuscript. Based on the reviews, we will not be able to accept this version of the manuscript, but we would be willing to review a much-revised version. We cannot, of course, promise publication at that time.

If you decide to revise the manuscript for further consideration at PLOS Genetics, please aim to resubmit within the next 60 days, unless it will take extra time to address the concerns of the reviewers, in which case we would appreciate an expected resubmission date by email to plosgenetics@plos.org.

[LINK]

We are sorry that we cannot be more positive about your manuscript at this stage. Please do not hesitate to contact us if you have any concerns or questions.

Yours sincerely,

Heather J Cordell

Associate Editor

PLOS Genetics

Gregory Barsh

Editor-in-Chief

PLOS Genetics

Reviewer's Responses to Questions

**Comments to the Authors:**

Reviewer #1: Review is uploaded as an attachment

Reviewer #2: This paper focuses on a topic that has attracted a lot of attention, i.e. the estimation of indirect effects in within family designs. The paper focuses on a specific type of indirect effects, i.e. indirect effects that originate in siblings rather than parents. It exploits data from both siblings and singletons to estimate such effects and provides a valuable addition to the field.

General comments

The authors justify their approach by a gain in power from including singletons, which seems fairly intuitive. However, because the estimation methods seem different (here test for a difference in association strength) is it worth justifying this point further either by developing the argumentation of why there is a gain in power (compared to which design) or by simulations?

Is it possible to include in the article the estimates of sibling effects from other designs (e.g. in adopted siblings or any comparable design using polygenic scores?) to verify whether effects are consistent with the approach used by the authors? At present, we have a suggested method but no benchmark either by simulation or by other designs showing whether this approach is reliable.

What about sex of the sibling? Does it make a difference whether sibling is same-sex or different sex If the authors interpret the findings as sibling influences on educational decisions, we could imagine that having a same-sex sibling going to uni would make more of a difference?

Comment on specific sections

Data from twin design etc. are not only usable “in principle”, they can and are used to estimate these effects. It is fair to say that these approaches are sensitive to modelling assumptions and measurement error, but so is the molecular genetic approach. So maybe present this as a complement to triangulate findings rather than an alternative.

I know that it is common in the UKB to adjust for 10 PCAs. The Kong paper adjusted for many more + interactions with birth year. Can the authors justify their choice?

It is interesting that the authors used the results from a within-sibship GWAS as sensitivity analysis. Please provide estimates using within-sibship GWAS weights in singletons and sibling families in addition to the attenuation. I think that the sentence “However, associations between this PGS and educational attainment in our sample (of unrelated individuals) may still be affected by these sources of association” requires a bit more explanation. The reader may assume that the point of within-sibship GWAS is to obtain SNP weights that are unaffected by these issues even when implemented in singleton samples (e.g. a SNP whose effect is entirely explained by pop strat or indirect effect would be given a weight of 0 and thus not enter the composition of the polygenic score). (Reading the discussion, it appears I misunderstood this first sentence as the authors do use the within-sibship GWAS to exclude these sources of bias as expected. Maybe the sentence can be clarified and developed to avoid misunderstanding. In particular, it may be good to mention more explicitly that the within-sibship GWAS controls for parental indirect effects but does still retain signal from sibling indirect effect. If it did not then we would expect findings to be radically different).

PGS associations and birth order

Sentence: “suggesting that the larger PGS association

in non-singletons is due to individuals with older siblings, rather than firstborns” makes sense but Figure 3 shows that PGS association is higher even in firstborns compared to non-singletons, so technically both would contribute to larger association in the larger non singleton sample. Maybe worth considering/discussing why firstborn and non-firstborn associations are higher than in the rest of the sample (e.g; selection effect) ?

“we found limited evidence for an interaction (P = 0.24)” I know that P values are not the best indicator, but this is what the authors are using here, and pvalue of 0.24 is “no evidence” rather than “limited”.

The whole approach relies on comparing the strengths of association in families with singletons vs non-singletons. And the difference is attributed to indirect genetic effects. This assumes that those families do not differ by any other characteristics, which may be a stretch. Same applies to the finding that the association is non-linear and different in families with 6+ children. There must be plenty of other factors that distinguish fairly rare families nowadays (6+ children) from others (e.g. religion, income available for each child). To what extent do these possible systematic differences affect the findings? The authors “lower investment or other confounders” in passing but not how these could affect estimates. (this is discussed more in the discussion section but may be worth flagging earlier in the intro or the results that this is an issue discussed at more lengths in methods and discussions).

Looking at Figure 4 makes me question the decision to fit a linear effect after removing 6 or more siblings, which seems post hoc and arbitrary. There seems to be a flattening already from 3 siblings. I think it is worth first fitting a model including both a quadratic and a linear term for number of siblings to check whether there is an overall significant pattern and only conduct linear analyses as follow-up analysis, restricting those analyses to the linear part of the fitted curve (or better, use the linear term in the quadratic model to estimate interaction with PGS). My intuition would be that such a linear term accounting for the quadratic trend should be larger than the one that the authors estimated).

One question that may not be pertinent. The analysis seems to suggest that nonsingeltons and non-firstborns (with not too many siblings) benefit from education-increasing indirect genetic effects via their sibling. How does it translate at the phenotypic level? Does it only make a difference in variances or also a difference of means (i.e. having a sibling helps with schooling). In an MR framework this association could be consistent with a causal effect of sibling education on the focal child’s education. I guess the effect would only increase the focal’s child education at the phenotypic level if the sibling had a higher polygenic load for education, and would decrease it if they had a lower polygenic load, leading to no expected differences in average between singletons and non-singletons.

Is it possible to provide the descriptives and formally test the differences in both variances and means as well as discuss what should we expect and what we observe at the phenotypic level. (Reading the rest of the results, I see that the authors provided some of the descriptives between singleton/nonsingletons and firstborns and nonfirstborns, including mean differences in education. I think it would be worth to also formally test for differences of variance in education and discuss these findings in a systematic way, i.e. spell out earlier in the text rather than in the discussion whether we expect differences in means and variances or not, and if yes, in which direction. And discuss whether findings are consistent or in contradiction to what would be expected under the indirect effect model?

Methods

“Indirect genetic effects of the maternal (), paternal ( ) and sibling ()

genotype on via the shared environment. These effects are captured by [ , ]”

Maybe “partially” captured? Cov[G1,Y] only captures indirect effects of transmitted rather than untransmitted parental alleles (i.e. )?

Linked to that, in the second equation we get 0.5Kns,m from KmGm in the first equation. Readers less familiar with these models may benefit from an explicit demonstration of this (i.e; comes from cov(G1,GM) being 0.5) as well as a more detailed derivation (maybe as supplementary). Also, not sure I get why *Var[G1] applies to the whole expression in brackets rather than only the first term (i.e. cov(G1,k1G1) = k1Var[G1]), which I could understand with the full derivation. This is important as the expression for the beta depends on Var[G1] applying to the full expression (although not sure it matters practically as variances of polygenic scores are unity).

“We constructed PGS for height, BMI and educational attainment by LD

Clumping (P < 1.0x10-5, r2 < 0.001, clumping distance = 10000 kb).” This is a fairly basic way of computing the polygenic score. Why choose this p-value, any justification? Why not choose a more advanced method to account for LD like LDpred? This would have the benefit of increasing the accuracy of polygenic scores.

Reviewer #3: In this paper, the authors use a creative design to try and estimate the magnitude of the sibling indirect effects in a PGS analysis for educational attainment, BMI, and height. Namely, they estimate the association of the PGS with its corresponding outcome in four subsamples of the UK Biobank: (a) those who report having no full siblings, (b) those who report having at least one sibling, (c) those who report having only younger siblings, and (d) those who report having at least one older sibling. Under the assumption that the direct and indirect genetic effects are the same across these different groups, a comparison of the PGS associations between these groups might shed light on whether a person's outcomes may be affected by the PGS of their sibling.

While I liked the idea behind the design of this study, I find myself very concerned about the limitations of this approach. I really appreciated that many of these limitations are brought up in a careful way in the discussion and methods section, though a bit more discussion on how important these limitations are would be useful. I describe these concerns in more detail below:

1. The authors show in the methods section how a key assumption of their design is that the direct genetic effects and indirect genetic effects of parents is the same in the different groups they consider. They state that one piece of evidence in support of their assumptions would be if the groups were similar on other observable characteristics. However, when they make comparisons across these different groups, they find significant differences in means on a number of variables. So how likely is it that their key assumption holds? One thing they may consider is to take the variables where they detect significant differences (e.g., men vs women) and test whether the PGS has a different coefficient when you divide the same that way. They may also consider other variables that are fixed early in life like birth coordinates.

2. The PGS the authors use are based on population-level GWAS that are blind to the family size of the individuals in the sample. If there is GxE on family size and the GWAS sample has more individuals with at least one sibling, then we would expect the PGS to be more predictive in a sample of people with siblings, wouldn't we. Given that most families have at least two children, it seems sensible that this could be a problem.

3. I was uncomfortable with the author's decision to drop estimates that correspond to families with 6 or more siblings. Once you acknowledge that environmental factors (like parental investment) may be influencing the patterns seen in the results, why couldn't it be that all the results are being driven by these factors?

4. In the methods section, they work out the math to infer how large the indirect effect from siblings under the assumptions of their method. Eyeballing figure 3, the difference between singletons and non-firstborns is about .07, suggesting the coefficient on the sibling PGS would be .14, almost as big as the association in singletons. Is this plausible?

5. Overall, I'd bring the assumptions and limitations much more front and center and expand the discussion of whether those assumptions are justified. And if they aren't justified, how large do you anticipate the biases would be as a result of the confounds.

**Have all data underlying the figures and results presented in the manuscript been provided?**

Reviewer #1: Yes

Reviewer #2: **No: **

Reviewer #3: Yes

PLOS authors have the option to publish the peer review history of their article (what does this mean?). If published, this will include your full peer review and any attached files.

Reviewer #1: No

Reviewer #2: No

Reviewer #3: No

---

## [Decision Letter · Decision Letter 1]

12 Jan 2022

Dear Dr Howe,

Thank you very much for submitting your Research Article entitled 'Evaluating indirect genetic effects of siblings using singletons' to PLOS Genetics.

The manuscript was fully evaluated at the editorial level and by independent peer reviewers. The reviewers appreciated the attention to an important problem, but one reviewer raised some substantial remaining concerns about the current version of the manuscript. Specifically, the reviewer points out that it is hard to disentangle your results from confounding due to gene-environment interactions. Given that there is existing evidence (albeit in a preprint) that goes against your results, the fear is that your findings could be largely explained by this confounding.

Based on these reviews, we will not be able to accept this version of the manuscript, but we would be willing to review a much-revised version. We cannot, of course, promise publication at that time.

Should you decide to revise the manuscript for further consideration here, your revisions should address the specific points made by the reviewer. We will also require a detailed list of your responses to the review comments and a description of the changes you have made in the manuscript.

If you decide to revise the manuscript for further consideration at PLOS Genetics, please aim to resubmit within the next 60 days, unless it will take extra time to address the concerns of the reviewers, in which case we would appreciate an expected resubmission date by email to plosgenetics@plos.org.

[LINK]

We are sorry that we cannot be more positive about your manuscript at this stage. Please do not hesitate to contact us if you have any concerns or questions.

Yours sincerely,

Heather J Cordell

Associate Editor

PLOS Genetics

Gregory Barsh

Editor-in-Chief

PLOS Genetics

Reviewer's Responses to Questions

**Comments to the Authors:**

Reviewer #1: The authors have thoroughly addressed my concerns. The revised manuscript is ready for publication in my opinion

Reviewer #2: The authors carefully answered my concerns.

Reviewer #3: The authors have made a large number of changes to the manuscript, highlighting how violations of the assumptions of their model affect their estimates and arguing why they believe their results are valid. Unfortunately, I still find their arguments unsatisfying. I describe my two main reasons below.

1. In their response letter (and the revised manuscript), the authors claim, "Group-level differences will only induce confounding bias if they influence both genotype and the outcome which is not possible for sex, age, and many other phenotypes which cannot plausibly influence germline genotype." While this may be true for each individual PGS association, it is not true for their test for indirect effects that the authors conduct in this paper. For example, if singletons are on average older, and older individuals have a weaker association between the PGS and EA, then you will detect a weaker EA/PGS association in singletons. Using the method proposed in this paper, the authors would then falsely conclude that there are significant indirect effects from siblings even if there are none. The authors acknowledge that "Family size is inversely associated with education-level because more education generally leads to having children later." This seems like a major unmeasured confounder of their test of sibling indirect effects. As a result, it is difficult to tell if the results reported in this paper are driven by a gene-environmental interactions or if they are driven by indirect effects of siblings.

One thing the authors may consider, which I described (admittedly poorly) in my previous review, is to look at whether there are significant difference in the PGS association for EA across the observed variables for which they see differences between singletons and non-singletons. For example, the authors report that the singletons in their data are more often male and are born earlier. What is the association between the PGS and EA in men vs women or between the older and younger individuals in their sample? If the associations are the same, this is some evidence that GxE is not driving their results, at least for the observed variables that they are able to test.

2. I appreciate the correction to Figure 3, but even a sibling indirect effect of .06 seems implausibly large to me. In response to Reviewer 1, the authors mention that Young et al. (2020) as an underpowered example of estimating indirect sibling effects. However, the more relevant paper is Kong et al. (2020), which uses a design where they regress EA on the proband, sibling, and (imputed) parental PGSs in the UK Biobank. The coefficient on the proband is .117 and the coefficient on the sibling is -.001 (SE=.013). This is quite precise and substantially smaller than the implied estimate from comparing singletons and non-singletons. Important, the data from Kong et al. is from the same dataset as the data used in this paper, so the environmental contexts should be similar. I find the estimates of Kong et al. much more reliable since they are direct estimates and don't require as strong assumptions. Can the authors justify why their results are different than those found in Kong et al.?

References:

Young, AI, Nehzati, SM, Lee, C., Benonisdottir, S., Cesarini, D., Benjamin, DJ, ... & Kong, A. (2020). Mendelian imputation of parental genotypes for genome-wide estimation of direct and indirect genetic effects. BioRxiv .

Kong, A., Benonisdottir, S., & Young, A. I. (2020). Family analysis with Mendelian imputations. BioRxiv.

**Have all data underlying the figures and results presented in the manuscript been provided?**

Reviewer #1: Yes

Reviewer #2: Yes

Reviewer #3: Yes

PLOS authors have the option to publish the peer review history of their article (what does this mean?). If published, this will include your full peer review and any attached files.

Reviewer #1: No

Reviewer #2: No

Reviewer #3: No

---

## [Decision Letter · Decision Letter 2]

21 Mar 2022

Dear Dr Howe,

Thank you very much for submitting your Research Article entitled 'Evaluating indirect genetic effects of siblings using singletons' to PLOS Genetics.

The manuscript was fully evaluated at the editorial level and by independent peer reviewers. The reviewers appreciated your improvements and revisions, but identified some remaining concerns that we ask you address in a revised manuscript. Specifically, as mentioned by one reviewer, a sentence in the discussion and a short supplementary note describing how your results compare to existing estimates using more direct methods (something along the lines of what you wrote in the response) would be valuable to readers.

We therefore ask you to modify the manuscript according to the review recommendations. 

[LINK]

Yours sincerely,

Heather J Cordell

Associate Editor

PLOS Genetics

Gregory Barsh

Editor-in-Chief

PLOS Genetics

Reviewer's Responses to Questions

**Comments to the Authors:**

Reviewer #1: I have no additional concerns and recommend the acceptance of the manuscript.

Reviewer #2: Manuscript ready for publication

Reviewer #3: I thank the authors for their clarifying comments I am substantially less concerned than I was before.

The point about the units of educational attainment is well taken. I'd note, however, that the PGS in Kong et al. is substantially more predictive than the PGS used in this paper. (I believe the implied R2 in this paper is roughly 1% while it is over 5% in Kong et al.) That said, if you assess the magnitude of the indirect sibling effect as a fraction of the population effect, I calculate a 95% confidence interval of (-11%,10%) for Kong et al. and (5%,15%), so they remain largely consistent even when accounting for differences in predictive power of the PGS.

If the editor agrees, I think a sentence in the discussion and a short supplemental note of how your estimates compare to more noisy estimates of indirect genetic effects which use sibling data would be valuable to readers.

**Have all data underlying the figures and results presented in the manuscript been provided?**

Reviewer #1: Yes

Reviewer #2: Yes

Reviewer #3: Yes

PLOS authors have the option to publish the peer review history of their article (what does this mean?). If published, this will include your full peer review and any attached files.

Reviewer #1: No

Reviewer #2: No

Reviewer #3: No

---

## [Editor Report · Decision Letter 3]

10 May 2022

Dear Dr Howe,

We are pleased to inform you that your manuscript entitled "Evaluating indirect genetic effects of siblings using singletons" has been editorially accepted for publication in PLOS Genetics. Congratulations!

Yours sincerely,

Heather J Cordell

Associate Editor

PLOS Genetics

Gregory Barsh

Editor-in-Chief

PLOS Genetics

Comments from the reviewers (if applicable):

**Data Deposition**

http://datadryad.org/submit?journalID=pgenetics&manu=PGENETICS-D-21-00943R3

**Press Queries**

---

## [Editor Report · Acceptance letter]

13 Jun 2022

PGENETICS-D-21-00943R3 

Evaluating indirect genetic effects of siblings using singletons 

Dear Dr Howe, 

We are pleased to inform you that your manuscript entitled "Evaluating indirect genetic effects of siblings using singletons" has been formally accepted for publication in PLOS Genetics! Your manuscript is now with our production department and you will be notified of the publication date in due course.

With kind regards,

Anita Estes

PLOS Genetics

On behalf of:
